# Neural Decoding of EEG Signals with Machine Learning: A Systematic Review

**DOI:** 10.3390/brainsci11111525

**Published:** 2021-11-18

**Authors:** Maham Saeidi, Waldemar Karwowski, Farzad V. Farahani, Krzysztof Fiok, Redha Taiar, P. A. Hancock, Awad Al-Juaid

**Affiliations:** 1Computational Neuroergonomics Laboratory, Department of Industrial Engineering and Management Systems, University of Central Florida, Orlando, FL 32816, USA; ffaraha2@jhu.edu (F.V.F.); fiok@ucf.edu (K.F.); 2Department of Biostatistics, Johns Hopkins University, Baltimore, MD 21218, USA; 3MATIM, Moulin de la Housse, Université de Reims Champagne Ardenne, CEDEX 02, 51687 Reims, France; redha.taiar@univ-reims.fr; 4Department of Psychology, University of Central Florida, Orlando, FL 32816, USA; peter.hancock@ucf.edu; 5Industrial Engineering Department, Taif University, Taif 26571, Saudi Arabia; amjuaid@tu.edu.sa

**Keywords:** brain signals classification, EEG, deep learning, machine learning, review

## Abstract

Electroencephalography (EEG) is a non-invasive technique used to record the brain’s evoked and induced electrical activity from the scalp. Artificial intelligence, particularly machine learning (ML) and deep learning (DL) algorithms, are increasingly being applied to EEG data for pattern analysis, group membership classification, and brain-computer interface purposes. This study aimed to systematically review recent advances in ML and DL supervised models for decoding and classifying EEG signals. Moreover, this article provides a comprehensive review of the state-of-the-art techniques used for EEG signal preprocessing and feature extraction. To this end, several academic databases were searched to explore relevant studies from the year 2000 to the present. Our results showed that the application of ML and DL in both mental workload and motor imagery tasks has received substantial attention in recent years. A total of 75% of DL studies applied convolutional neural networks with various learning algorithms, and 36% of ML studies achieved competitive accuracy by using a support vector machine algorithm. Wavelet transform was found to be the most common feature extraction method used for all types of tasks. We further examined the specific feature extraction methods and end classifier recommendations discovered in this systematic review.

## 1. Introduction

The human brain is a complex system containing approximately 100 billion neurons and trillions of synaptic connections [1,2]. The brain’s electrical activity became a research focus in the 19th century when Richard Caton recorded brain signals from rabbits [3,4]. Brain recordings were also performed by Hans Berger, the first person to record electroencephalogram (EEG) signals from the human scalp [5]. EEG-based research has since increased significantly, and EEG is now the most commonly used noninvasive tool to study dynamic signatures in the human brain [6,7]. EEG signals measure voltage fluctuations at the scalp and reflect the instantaneous superposition of electric dipoles, primarily from dendritic inputs to large pyramidal cells in the neuropil [8]. Signals traveling in white matter have traditionally been thought to be too fast to superimpose temporally, although recent cable theoretic models [9] and empirical work [10] suggest that white matter may also contribute to brain rhythms measured at the scalp. Classically, the three primary forms of the brain’s activity based on EEG signals are brain waves, event-related potential (ERP), and steady-state visual evoked potentials (SSVEPs). Among those, brain waves are most commonly used in EEG signal analysis for different types of tasks. Brain waves have been categorized in terms of five frequency bands: delta, 0.5–4 Hz; theta, 4–8 Hz; alpha, 8–13 Hz; beta, 13–30 Hz; and gamma, 30–150 Hz [11]. Other classifications of brain signals can be found in previous publications [12,13].

EEG is a low cost, noninvasive neuroimaging technique that provides high temporal resolution recordings of dynamic signatures in the brain; it has therefore become an indispensable tool in a variety of applications, including clinical diagnosis of a range of epileptic seizure types [14,15], brain-computer interface (BCI) systems, sleep analysis, and decoding behavioral activity [6,16,17]. When interpreted carefully [18], classification tools can be used not only for prediction but also to gain neuroscientific knowledge. However, EEG signals are complex, high-dimensional [19] and non-stationary, and have a low signal-to-noise ratio in the temporal domain. Therefore, careful preprocessing is often required to remove artifacts [20], particularly when EEG data are collected concurrently and in the MRI setting [21].

Although EEG has been demonstrated to be a valuable tool for research in various applications, it has several limitations, such as a low signal-to-noise ratio [22,23], nonlinearity and nonstationary properties [24,25], and inter-individual variability [26] which affect analysis and processing performance. To address these limitations, EEG signal processing pipelines are often used. The general EEG classification pipeline includes data preprocessing, initializing the classification procedure, splitting the data set for the classifier, predicting the class of new data, and evaluating the classification model for the test data set [27]. As shown in references [28,29], Riemannian geometry-based classifiers and adaptive classifiers have achieved success in classifying EEG signals. Machine learning (ML) and deep learning (DL) methods have become rapidly growing areas with applications in computational neuroscience, owing to higher levels of neural data analysis efficiency and decoding brain function [30]. In ML and DL, various algorithms are used simplify processing pipelines and improve the learning process. For instance, supervised ML algorithms first learn on training data. The model and learned parameters are then applied to unseen or new data to predict the class label of the new data [27]. Among different types of classification tasks, binary and multi-label classifications are widely used in clinical studies, and in studies of cognitive function, motor imagery (MI) processing, emotion recognition (ER), and brain disorders, including brain injury, attention disorders, and multiple sclerosis [31,32,33,34]. In addition to these recognized areas, various computational neural models have been considered, such as the medial prefrontal cortex (mPFC) and anterior cingulate cortex in modeling of learning predictions and monitoring behavior [35]. A recent computational model, the predicted response-outcome model [36] is based on the hypothesis that mPFC stores predictions of future outcomes. According to this model, Garofalo et al. [35] have proposed that mediofrontal ERP signals of prediction errors are modulated by the likelihood of occurrence during a task. Furthermore, reference [37] has identified a causal role of the ventromedial prefrontal cortex (vmPFC) in the acquisition of fear conditioning in the definition of stimulus-outcome contingencies.

Because of their outstanding robustness and adaptability, several ML and DL models for performing EEG signal classification have been reported. The primary purpose of this study was to review and explore the recent advances over the past two decades in the deployment of supervised ML algorithms as well as DL models for the classification of EEG signals. We investigated the overall trends and ML and DL models used in individual studies and compared classification algorithms based on different tasks. We propose six different categories of tasks on the basis of the studies reviewed herein. We attempted to address defined research questions concerning each of these categories. According to this evaluation, we provide recommendations for choosing a suitable classifier for use in future applications. The remaining sections of this manuscript are organized as follows. In Section 2, we describe the method used to identify and select publications for subsequent inclusion in our synthesis of research studies. In Section 3, we describe general steps to generate an EEG-based BCI system. We review the typical pipeline used to construct EEG classifications, including data acquisition, artifact removal, feature extraction, and classification. In Section 4, we provide a general overview of the literature search, study characteristics, validity assessments, and the main findings of each of the articles reviewed. In Section 5, we discuss applications of ML and DL techniques for different types of tasks, and provide recommendations regarding the selection of an effective classifier for each task. Finally, Section 6 highlights future perspectives in this field.

## 2. Materials and Methods

Preferred Reporting Items for Systematic Reviews and Meta-Analyses (PRISMA), a standard systematic review and meta-analysis guideline [38], was used in this study. An important component of this systematic review involved the clear definition of research questions to reduce the effects of research expectations. Furthermore, our research method followed the Cochrane Collaboration definitions [39] to minimize the risk of bias.

### 2.1. Research Questions (RQs)

RQ1: What classification tasks have received the most attention with the introduction of ML and DL algorithms and the use of EEG brain data?RQ2: Which feature extraction methods were used for each task to extract appropriate inputs for ML and DL classifiers?RQ3: What are the most frequently used ML and DL algorithms for EEG data processing?RQ4: Which specific ML and DL models are suitable for classifying EEG data involving different types of tasks?

### 2.2. Search Strategy

In the exploration phase, we used a search strategy with an organized structure of key terms used to perform comprehensive literature searches in databases. These keywords played a key role in identifying relevant studies and helped us focus on specific publications addressing the research questions. The literature search used multiple academic databases, including Web of Science, IEEE Xplore, Science Direct, arXiv, and PubMed, and the following groups of keywords in the article titles, keywords or abstracts: (“Machine Learning” OR “Deep Learning” OR “Classification” OR “Supervised Learning” OR “Neural Networks”) AND (“Electroencephalography” OR “EEG”).

### 2.3. Criteria for Identification of Studies

After searching the databases, we established a protocol based on specific inclusion and exclusion criteria to identify publications qualified for inclusion in our review. The eligible publications met the following inclusion criteria: (a) publication in English; (b) inclusion of EEG data; (c) application to the classification of brain activities in humans; (d) publication in a peer-reviewed journal; and (e) publication within the past 21 years (2000–2021). The exclusion criteria used during the screening process included: (a) publication in a non-peer-reviewed journal; (b) content types of dissertations, published abstracts, or book chapters; and (c) experimental studies performed in infants. The relevant studies were selected on the basis of the inclusion and exclusion criteria and several data categories were collected including task information (i.e., task type, number of subjects, and number of EEG channels used), databases used, frequency range used for analysis, feature extraction methods, ML/DL algorithms, and their performance (see Table A1 in Appendix A, for details).

## 3. Theoretical Background

Figure 1 illustrates the general pipeline for the construction of a classification model based on EEG data. Processes including data acquisition, preprocessing, feature extraction, and ML and DL models can be differentiated from one another. In the following, we review the existing computational methods for each step.

### 3.1. EEG Data Acquisition

Typically, EEG recordings were obtained by using an international 10–20 or 10–5 electrode placement system introduced by the American EEG Society. In this method, multiple noninvasive electrodes are placed on the surface of the scalp. Each electrode receives brain signals, which are then amplified and sent to a computer after being converted them into line images representing brain waves. Electrodes used in EEG recording can be categorized as wet electrodes, semi-dry electrodes, and dry electrodes [40]. In wet electrode recording, conductive gel or paste, which is very uncomfortable for participants, must be placed between the electrodes and the skin. Semi-dry electrodes require a small amount of conductive gel, and dry electrodes do not require conductive gel or skin preparation [41,42]. Despite its ease of use and quick setup, EEG recording from dry electrodes yields data with more artifacts than does gel-based EEG recording, thus affecting EEG analysis. In this case, identifying the most sensitive biomarkers for the specific task is essential [43]. In addition, some experimental studies have confirmed that selecting the proper features can improve the performance of ML/DL methods. For instance, Dehais et al. [44] have analyzed the performance of gel-free EEG recording with ERP and power spectral density (PSD) features along with the linear discriminant analysis (LDA) method and found that the performance of LDA with PSD features is much better than that of LDA with ERP. Thus, ERP appears to be more sensitive to noise than PSD features.

### 3.2. Artifacts in EEG Signals and Preprocessing

The preprocessing step facilitates the removal of low-quality data without altering the clean data. This process also fragments the continuous raw signals without changing the data [45]. Artifact removal is an essential preprocessing step in the analysis of EEG signals, because the recordings usually include a significant number of extrinsic artifacts associated with environmental noise and experimental error, as well as intrinsic biological artifacts associated with body function (e.g., eye blinking, movement, respiration, and heartbeat) [46]. Various simple methods can be used to eliminate non-biological artifacts from EEG signals. Because environmental artifacts do not have the same frequency as the EEG signals of interest, they can be eliminated through application of a band filter [46]. Alternatively, standard operating procedures provide proper operational guidance for the data acquisition step and decreasing experimental artifacts [47]. However, the main types of biological noise include ocular artifacts, muscle artifacts, cardiac artifacts, and instrument artifacts, which require the use of filtering and/or computational methods to be removed from EEG data. More in-depth descriptions of these types of artifacts can be found in references [46,48]. According to the literature, extensive research has been conducted in the past decades [46,49] to identify and define efficient methods for both automatic and manual artifact removal.

#### 3.2.1. Regression Methods

A typical approach to artifact removal is the regression method [50], which removes estimated artifacts by determining the amplitude relation of the reference. Hence, signals such as those from an electrocardiogram (ECG) or electrooculogram (EOG) are required to separate artifacts from EEG signals. Although the regression method is based on simple mathematical intuition and therefore is widely used because of the minimal computation required, the dependence of this method on reference channels for ECG and EOG removal is considered a drawback [48].

#### 3.2.2. Blind Source Separation Methods

Blind source separation (BSS) methods are based on the hypothesis that a combination of several distinct original signals results in the signals observed on a multi-channel recording; thus, neither more reference channels nor prior information is required [51]. Three typical methods using the BSS algorithm are principal component analysis (PCA), independent component analysis (ICA), and canonical correlation analysis (CCA). ICA [52,53] is a statistical algorithm that is used for solving the BSS problem and considers random variables to maximize the independence of the output components via the discovery of a linear transformation. Furthermore, ICA is a powerful tool that reduces dimensions and extracts independent components from original signals. This method has good performance in extracting artifacts such as eye blinks and heartbeats, because they are generated by independent sources and are not associated with particular frequencies. ICA is efficient according to the assumption that original signals are statistically independent of each other and have a non-Gaussian distribution. Moreover, the signal dimension must be greater than the source signal. PCA [54,55] maximizes the variance of the transformed data and depends only on the second-order statistics of covariance [51]. PCA is a well-known method to reduce the dimensionality of the features while protecting their statistical information. The advantage of this method is that it retains the variance of the data set. However, its disadvantage is that if the potentials of drifts and EEG data are similar, PCA cannot extract the appropriate interferences. CCA is widely used in SSVEP-based brain BCI to identify the frequency components of EEG that characterize visual stimulus frequencies [56]. Of note, a comparison of BSS methods is necessary, but that topic is beyond the scope of this review article. Reference [57] provides further information about these methods.

#### 3.2.3. Wavelet Transform

Wavelet transform (WT) [58,59,60,61] is a spectral estimation method that converts a time-domain signal into a time and frequency domain signal. After decomposition of wavelet transformation on EEG signals and during artifact removal, WT localizes the features and maintains them during the filtering process by defining a threshold for the elimination of noise signals. Whereas WT has good performance in analyzing the components of non-stationary signals, it fails to recognize artifacts that overlap with the spectral properties. Accordingly, new hybrid methods such as wavelet-BSS have been proposed to overcome this drawback [62].

#### 3.2.4. Filtering Methods

Various filtering approaches that have been used for EEG artifacts and noise-canceling include frequency filtering, adaptive filtering, and Wiener filtering [63].

##### Frequency Filtering

Frequency filtering is a simple classical separation technique to eliminate artifacts from the desired EEG signals. According to Hu and Zhang [45], four types of frequency filters can be considered: low-pass, high-pass, band-pass, and band-stop filters [45]. However, this method is not efficient if the spectral distributions of artifacts and the EEG components overlap. In the case of overlap, alternative artifact removal techniques are necessary [63].

##### Adaptive Filtering

Adaptive filtering is based on the assumption that the EEG signal of interest and the artifact are uncorrelated. This filter uses a reference signal and produces an estimated signal that is correlated with the artifact; the estimate is then subtracted from the primary signals to yield a noise-free EEG signal [64]. Adaptive filtering uses the least mean squares (LMS) algorithm, which is linear in convergence, to assess the clean signals by upgrading the weight parameter. Another optimization algorithm, the recursive least squares algorithm, is quadratic in convergence and is an extension of LMS [48]. Depending on the type of recursive least squares algorithm, its convergence may be faster than that of the LMS algorithm, but its computational cost is greater. A disadvantage of using adaptive filtering is that providing reference input requires more sensors [64].

##### Wiener Filtering

Wiener filtering is a statistical technique to minimize the mean square error between the signals of interest and the estimated signals by generating a linear time invariant filter [63]. Although Wiener filtering does not require an additional reference, because the minimization process is applied to estimate the power spectral densities of the EEG signal and artifact signal, the computational process can be complicated. In addition to the listed techniques, many other efficient methods exist, such as CCA, empirical mode decomposition (EMD), and sparse decomposition methods. Furthermore, hybrid methods combining these preprocessing algorithms and other methods such as EMD-BSS, wavelet-BSS, and others, have been used to maximize the efficiency of the algorithm [65,66]. Further details can be found in previous publications [46,48].

### 3.3. Feature Extraction Methods

EEG signals are typically complicated and contain a large amount of information. Thus, the ability to extract the proper features from EEG signals is a critical component of any successful ML and DL algorithm. The feature extraction step aims to transform the data into a low-dimensional space while maintaining the critical information conveyed by the EEG signals [67]. According to the literature, many feature extraction methods have been proposed on the basis of the specific tasks, including time domain, frequency domain, and time-frequency domain as well as spatial information in the signals [68,69]. Among these extraction methods, ICA, PCA, and autoregressive (AR) models are considered time-domain methods. Statistical measures to evaluate parameters such as the mean, standard deviation, variance, root mean square, skewness, kurtosis, relative band energy, and entropy also fall into this category [70]. Fast Fourier transform (FFT) and Welch’s method [71] are among the frequency-domain methods used to analyze EEG signals. WT and short time Fourier transform (STFT) are two standard time-frequency domain methods that extract features based on both time and frequency. Findings from recent studies have demonstrated the advantages and drawbacks of each feature extraction method, and have indicated that care must be taken in selecting the appropriate method for a specific type of task [67]. Here, we present a general overview of the most commonly used methods for analyzing EEG signals. More details on the feature extraction methods used according to the specific nature of the task are provided in the sections below.

#### 3.3.1. Principal Component Analysis

PCA is a linear transformation that is widely used to reduce dimensionality. PCA introduces a vector in a lower-dimensional space to reduce signal complexity over time and space [72]. Although PCA is used to separate artifacts from original signals, this transformation can also be used for feature extraction without information loss [70]. PCA creates a set of linear vectors that are not correlated with one another (i.e., principal components) via converting correlated variables from original signals [73]. Although principal components enhance signal similarity and improve the effectiveness of classification of data [74], they are not as interpretable as primary features. In addition, PCA does not work well in analysis of complex data sets [75]. To address these drawbacks, several variation in PCA have been proposed in EEG data analysis such as kernel PCA [76] or sparse PCA [77].

#### 3.3.2. Autoregressive Model

The AR model is a feature extraction method for frequency domain analysis, which has been used to analyze non-stationary signals such as EEG data [73]. AR assumes that real EEG signals can be predicted by the AR process; this prediction can be performed with the order and parameters of the approximation model. The AR model’s order is a value from 1 to 12, which effectively indicates the performance of the model. However, selecting an appropriate value for the order of the AR can be challenging, because improper order selection causes miscomputed spectrum estimation, which may increase the computational costs [78].

AR methods such as bilinear AAR, adaptive AR parameters, and multivariate AAR have been widely used in EEG data analysis, thus allowing the AR model parameters to adapt to nonstationary EEG signals [79]. These methods help provide successive parameter estimation and minimize prediction error. For example, AR parameters can be evaluated in an adaptive AR model by using the Kalman filter, thus improving classification performance up to 83% [74]. Other advantages of the AR model include that it provides appropriate frequency resolution [75] and can be used estimate the power spectra of shorter segments of EEG data in various applications [80]. However, AR is vulnerable to inappropriate order and parameter selection [75].

#### 3.3.3. Fast Fourier Transform

FFT is an effective method for stationary signals. It transforms signals from the time domain to the frequency domain and implements spectral analysis [67]. In this method, features are extracted by using mathematical tools to calculate the PSD. The estimation of PSD for a related band can be computed with FFT, which uses non-parametric methods such as Welch’s method [67,81]. Although FFT is commonly used in the data analysis process, and it works effectively for stationary signals, it is not efficient for nonlinearity and nonstationary data such as EEG signals; moreover, the results obtained through this method are not reliable. This shortcoming has motivated researchers to develop novel procedures and methods for the analysis of nonstationary signals, such as the Fourier decomposition method [82], variational mode decomposition (VMD) method [83], and Hilbert-Huang transform (HHT) method [84].

#### 3.3.4. Wavelet Transform

WT is a time-frequency transform that considers the features of the EEG signals within a frequency domain and is perfectly localized within the time domain [70]. This method has good performance in spectral analysis of irregular and nonstationary signals within different size windows [85]. One advantage of WT is that it provides accurate frequency information and time information at low and high frequencies, respectively. That is, a narrow window is typically used to evaluate high frequencies, and a wide window is applied to assess low frequencies [68,86]. Thus, WT is suited for transient oscillation in signals, particularly biosignal data, which consist of low-frequency components with long-time periods and high-frequency components with short-time periods [74]. However, WT suffers from Heisenberg uncertainty, which negatively affects its performance [74]. WT evaluates small wavelets within a specific range for a limited duration. The wavelets have oscillating motion starting at zero; these oscillations increase and then decrease to zero [87].

#### 3.3.5. Common Spatial Pattern

CSP is a successful feature extraction method in BCI applications, particularly motor imagery tasks [88], that can be used for spatial filtering by using the whole data trail or by splitting trails into time segments. CSP is widely used for binary classification tasks [89]. The objective of CSP is to differentiate between classes by minimizing the variance of one class and maximizing the variance of the other class. This process can be implemented by introducing spatial filters for each class. With this method, EEG signals are transformed into a variance matrix representing the discrimination between classes [73].

The main advantage of using CSP is that it is simple and can be executed rapidly [73]. However, this method has some inherent limitations in extracting optimal features from raw EEG data [88]. To address this issue, several studies have proposed optimal spatial feature selection methods. For example, Jin et al. [88] have developed a novel feature selection method based on an improved objective function by using Dempster-Shafer theory, considering feature distribution. Furthermore, CSP is highly sensitive to artifacts present in the original data set, and changing the positions of electrodes affects the classification accuracy [90]. According to reference [91], several parameters such as the frequency band filter, the time segment, and the subset of CSP filters to be used should be considered to have an effective CSP algorithm. Thus, the performance of the CSP algorithm depends on the subject-specific frequency band. Many approaches have been proposed to address the issue of identifying the optimal frequency band for the CSP algorithms, such as the common spatio-spectral pattern (CSSP), common sparse spectral-spatial pattern (CSSSP), spectrally weighted common spatial pattern, sub-band common spatial pattern, and discriminant filter bank common spatial pattern [92,93]. CSSP uses a simple time delay embedding with the CSP algorithm, which improves the CSP’s performance by optimizing the frequency band at each electrode position [94]. However, the non-stationary EEG data decrease the performance of the CSSP. To overcome this challenge, Cho et al. [95] have included a noise removal term in the Rayleigh coefficient of CSSP and have designed an invariant CSSP algorithm that is both consistent and robust to noise. However, the drawback of the invariant CSSP is that the optimal noise removal value must be determined. The CSSSP [96] has been proposed to enhance the performance of CSSP.

In contrast to the CSSP, which identifies various spectral patterns for each channel, this algorithm searches for a common spectral pattern for all channels [93]. Sub-band common spatial pattern [97] is an other extension of CSP used to filter EEG signals at multiple sub-bands to extract CSP features from each sub-bands, regardless of the associations among features from different sub-bands [93]. To overcome this limitation, a discriminant filter bank common spatial pattern [98] that uses the Fisher ratio of single channel band power values has also been proposed. Further information regarding the application range of the CSP can be obtained in reference [93].

### 3.4. Classification Algorithms

Artificial intelligence includes ML, and DL is a rapidly growing area with applications in the classification process [99]. The objective of classification is to predict the class label of the new data points in various tasks [27]. Classification algorithms can be divided into two categories: conventional classification algorithms and DL algorithms [100]. Conventional classification algorithms represent a precise effort to build classification models by using input data, and applying statistical analysis to classify output values. Most conventional classification algorithms use hand-crafted input features to train the model. This process, called feature creation, has limitations in handling input in high dimensional data sets [100]. DL algorithms rely on representation learning [101] and can accommodate the limitations of conventional classification algorithms through learning features automatically at multiple levels of abstraction [100]. Table 1 presents a brief comparison of conventional classification and DL algorithms [100].

#### 3.4.1. Conventional Classification Algorithms

Among different conventional classification algorithms including supervised learning and unsupervised learning, supervised algorithms are the most well-known methods used in EEG data analysis [102]. One of the most commonly used supervised algorithms is artificial neural networks (ANNs). ANNs are computational models [103] that use multi-layered networks of neurons with weighted connections between units, typically followed by a static non-linearity function (e.g., ReLu). During the learning phase, the network can learn by modifying its weights to enhance the performance outcomes in test data classification [104]. Similar examples of well-performed and well-known supervised algorithms include naive Bayes (NB), support vector machine (SVM), k-nearest neighbor (KNN), logistic regression (LR), random forest (RF), and LDA. Each supervised model applies a learning algorithm to generate a more accurate model [105].

NB is a probabilistic classifier that applies Bayes’ theorem to classify data on the basis of certain features [106]. It is a simple and effective classifier that needs only small training data sets to estimate the parameters for classification. This advantage makes NB a robust classifier for EEG signals analysis in several types of tasks such as ER [107], seizure detection (SD) [108], and MI [109]. However, NB is based on the assumption that all attributes are independent of one another, and feature vectors have equal effects on the outcome [106].

SVM has been demonstrated to be a useful supervised model based on a statistical learning tool with high generalization. The principle underlying SVM is the separation of two data sets. This separation can be linear or non-linear. In the case of linear separation, SVM uses a discriminant hyperplane to distinguish classes. However, in the case of non-linear separation, SVM uses the kernel function to identify decision boundaries. Compared with that of other supervised algorithms, such as ANNs and KNN, the computational complexity of SVM is low [110,111]. Although the computational complexity of KNN decreases by increasing the k-value, its classification performance also decreases [110,112]. Furthermore, with the advent of DL algorithms, SVM has remained widely used in EEG signal classification, because its computation has a solid mathematical basis. However, the performance of SVM is affected by the kernel function and penalty coefficient parameters; thus, optimizing the parameters introduced into SVM classifiers is essential [113]. Huange et al. [114] have applied a genetic algorithm, and Wang et al. [115] have proposed particle swarm optimization to optimize SVM parameters. According to our investigation, SVM has been widely used in EEG signal classification because of its simplicity and adaptability in solving classification problems such as diagnosis of brain disorders (e.g., SD, and Alzheimer’s disease) [116,117,118].

RF is a tree-based supervised algorithm that constructs an ensemble of decision trees. Each decision tree is generated during the training phase. RF makes predictions from each tree and selects the final decision via a voting method or averaging the results [119] to identify the most commonly used class. The main idea underlying this and related ensemble methods is that a group of weak classifiers can collectively generate a strong classifier to create a successful learning algorithm. However, the overfitting and instability of trees can affect RF model performance, particularly with varying sizes of trees [106]. In contrast to the LR model, which is a probabilistic classification model for both binary and multi-class classification tasks [120], RF works on both discrete and continuous data, thus providing models for classification and regression problems. Furthermore, the parallelization structure of RF results in better performance than that of the other supervised algorithms on large EEG data sets in addressing classification problems [109].

LDA is a linear transformation technique used to identify linear combinations of the variables that most effectively separate the classes [121,122]. LDA is based on the assumption that the density for the data is normally distributed, with equal covariance for both classes. The separating hyperplane is achieved by maximizing the distance between the two classes, while minimizes the distance points within each class [123]. This technique is simple to use and has very low computational requirements. Consequently, LDA has been successfully applied to address classification problems in BCI systems such as MI based BCI [124], P300 speller [125], and multiclass BCI [126]. However, the main limitation of this model is its linear nature which prevents competitive results on nonlinear EEG data [127,128].

#### 3.4.2. Deep Learning Algorithms

DL is a new branch of ML that has received widespread attention in EEG classification tasks. Although conventional classification algorithms have been very effective in analyzing massive data sets and understanding the relationship between variables, such algorithms often lead to poor generalization behavior and low classification performance when highly dynamic features are encountered [129]. DL algorithms inspired by neuroscience [130] exploit learning features from raw data without depending completely on preprocessing [131]. Humans can transfer knowledge and memory throughout their lifespan, whereas DL algorithms immediately forget the previous learning after being trained on a new data set [132]. For instance, Borgomaneri et al. [133] have confirmed that fear memory remains in humans after memory reactivation and affects the future learning process. Such a life-long learning task is a large challenge in developing neural network algorithms [134]. Furthermore, DL algorithms apply multiple layers of perceptrons that obtain representation learning [131]. Recent developments in graphics processing unit technology have enabled the development of DL architectures on large data sets. This advantage has significantly improved the performance on large data sets with high-dimensional data [20].

A convolutional neural network (CNN) is a type of deep neural network that has gained attention, particularly in computer vision and neuroimaging [131]. CNN can identify the image of an object by using convolutions within its architecture; including convolutional layers that have parameters to create a feature map; pooling layers that reduce the number of features for computational efficiency; dropout layers that help avoid overfitting by randomly turning off perceptrons; and a output layer that map the learned features into the final decision, such as classification [135,136]. The recent emergence of the CNN algorithm has enabled outstanding performance in several application such as image processing, natural language processing, and classification of EEG recordings, particularly for MI tasks [137,138,139,140]. However, CNN performance is highly dependent on hyperparameters such as the number of convolution layers, and the size and number of kernels and pooling windows [137]. Fortunately, CNN architectures have led to automatic optimization of parameters through several iterations [20]; therefore, CNN is very commonly used for addressing classification problems involving large data sets.

A recurrent neural network (RNN) is a time series-based DL algorithm that uses sequential data and learns from training data, similar to feed-forward and CNN methods. Unlike traditional deep neural networks, which assume independence of the input and output, RNN takes information from inputs continually. Consequently the output of RNN depends on the prior outputs and the current inputs within the sequence [137]. Thus, the architecture of this network includes inbuilt memory cells for storing information from previous output states [141]. The form of RNN architecture has enabled these types of networks to effectively analyze time series data for applications such as speech recognition [142], natural language processing [143], and disease signal identification [144]. The most commonly used RNN variants are long short-term memory (LSTM) [145], LSTM peephole connections [146], gated recurrent units [147], and multiplicative LSTM [148]. The ability of these variants to preserve and retrieve memories is part of the generic structure of these networks. However, CNNs have a different architecture and use filters and pooling layers. According to differences in the internal network structure of CNNs and RNNs, CNNs are effective for analysis of spatial data, because CNN models consider the complete trial as an object and can extract features. In contrast, RNN models are suited for analysis of temporal and sequential data by slicing the trail into several subtrails [137].

## 4. Results

### 4.1. Literature Search

Figure 2 is a flow diagram of the study selection process based on the PRISMA guidelines [38]. As shown in the diagram, we obtained 764 articles from all database searches after the removal of duplicates. To determine which articles were appropriate to consider in this study, we reviewed all abstracts to determine whether the findings met our inclusion criteria. A total of 254 articles remained after this evaluation. The full text of each of these 254 articles was reviewed, and 128 relevant articles met all the aforementioned criteria. The selected studies included 78 articles published in journals, 46 conference and symposium articles, and 4 preprints. To improve understanding of the evolution of research in this domain, we also considered the temporal distribution of these publications (Figure 3). Since 2016, the importance of supervised ML and DL classifications and their role in analyzing EEG data has received increasing attention from the research community; more than 71% (91 of 128) of the selected studies were published during this period (i.e., 2016–2021). The remaining studies (29%) were published between 2000 and 2015. We anticipate that the number of publications in this domain will continue to grow substantially over the next few years.

### 4.2. Quality Assessment

To assess the quality of the selected articles, we applied the Cochrane Collaboration tool [39]. According to this tool, the articles were identified as having an (a) low risk of bias, (b) high risk of bias, or (c) unclear risk of bias. The overall quality of the articles was categorized as weak (fewer than three low-risk domains), fair (three low-risk domains), or good (more than three low-risk domains). Of the 128 studies considered in this systematic review, 40 articles were categorized as good, 26 articles were categorized as fair, and 62 articles were categorized as weak (Figure 4).

### 4.3. Study Characteristics

The tasks presented were in studies that were organized into six groups: ER (15%), mental workload (MWL) (18%), MI (20%), SD (19%), sleep stage scoring (SS) (7%), and diagnosis of neurodegenerative diseases (ND), including Alzheimer’s disease (AD), Parkinson’s disease (PD), and schizophrenia (SZ) (9%). Other studies (12%) focused on ERP [47,149,150,151,152], anxiety and stress [153,154], depression [33,66,155], the detection of alcoholism [156], auditory diseases [157], attention deficit hyperactivity disorder [158], sleep apnea [159], and the classification of creativity [160]. In recent years, the application of supervised ML and DL models in MI and ER has gained significant attention, despite decades-long research studies already in progress in these fields (Figure 5).

In the selected articles, two types of data sets have been used: (1) experimental data sets associated with a specific project or research direction and (2) freely available data sets comprising EEG data associated with various tasks. In the first type of data set, EEG recordings are obtained from participants by using non-invasive electrodes placed on the scalp surface. Of the 128 studies included in this systematic review, 66 studies (52%) generated experimental data sets; the remaining studies used data in publicly maintained databases. Table 2 includes a list of the public data sets frequently used for various tasks in the selected studies. The descriptions include the numbers of participants and the target tasks. Corresponding to the type of data set, the number of participants varied among studies depending on the type of task involved (Figure 6).

### 4.4. Which Feature Extraction Methods Were Used for Each Task to Extract Appropriate Inputs for Machine Learning and Deep Learning Classifiers?

When using a feature extraction method, the principal objective is to minimize the loss of essential characteristics embedded in the EEG signals. In addition, feature extraction methods provide conditions that promote the optimal selection of features that are important to the specific classification task. Among the articles reviewed for this study, various methods have been used to extract the features from EEG signals associated with different type of tasks. Figure 7 shows a plot of the feature extraction methods used for each of the six categories: ER, MWL, MI, SD, SS, and ND tasks. The inner circle represents the type of task, and the outer circle represents the utilization rate of the method for each task. The results suggested that EEG signals were commonly analyzed in the time-frequency domain in the reviewed articles. WT was the most common feature extraction method used for all tasks. Our investigation revealed that researchers examining MWL, ND, and SS tasks typically chose FFT; in contrast, PCA was heavily used for feature extraction in studies focused on ER, MWL, and SD tasks. Furthermore, the CSP model was a commonly used feature extraction method in EEG-based BCI systems for MI tasks. In some research studies, the combination of CSP and other methods was used to decode the EEG signals. For example, Wang et al. [208] have examined the performance of CSP with and without discriminative canonical pattern matching method and have found that a combination of CSP and discriminative canonical pattern matching, as compared with the single CSP method, significantly improved the accuracy of feature extraction in pre-movement EEG patterns by 10%. Several other efficient feature extraction methods have been used in studies involving various tasks, such as Higher-Order Spectra (HOS) for the PD detection task [209], one-dimensional ternary patterns (1D-TP) for the SD task [187], and Kolmogorov complexity (Kc) and sparse spectrotemporal decomposition (SSD) for the MI task [210].

### 4.5. What Are the Most Frequently Used Machine Learning and Deep Learning Algorithms for EEG Data Processing?

Among the studies performed over the past 21 years that have focused on classifying EEG signals, SVM was the preferred supervised algorithm for nearly all types of tasks; more than 40% of studies achieved the highest accuracy by using this ML classifier. Figure 8 indicates the plot of ML and DL classification algorithms used for each of the six types of tasks: ER, MWL, MI, SD, SS, and ND. The inner circle represents the tasks, and the outer circle represents the utilization rate of the classification algorithms for each task. As confirmed in Figure 8, CNN, KNN, and RF were among the most accurate classifiers after the SVM algorithm.

Assessing the performance of classification models depends on the type of data set (i.e., size and quality) as well as selected feature extraction method. To obtain more intuitive information about our findings, we plotted a bubble chart on a 2-dimensional axis showing the relationships between ML and DL algorithms with feature extraction methods (Figure 9). The size of each bubble shows the performance of classification models for each task, as marked with different colors. As Figure 9 demonstrates, the use of SVM resulted in a wide range of performance in tasks including ER, MWL, and SD, when frequency-domain and time-frequency domain feature extraction methods, such as FFT and WT were chosen. RF had competitive performance in tasks that included ER, MWL, MI, SD, and SS, particularly in references [59,104,119,185,211]. However, in the domain associated with ND tasks, the RF algorithm did not perform as well as in other tasks. According to two recent studies [209,212], the best performance in the ND task classification was achieved when SVM and KNN were applied. More interestingly, none of the MI studies reported high classification accuracy with the KNN classifier; instead, LDA resulted in high performance in only the MI and SD tasks. Furthermore, classification algorithms known as AdaBoost achieved the highest classification accuracy rate for the SS task when the EMD feature extraction method was chosen. DL algorithms, particularly CNN, showed promising performance in the ER, MWL, and MI tasks.

## 5. Discussion

An important EEG signal is the P300 ERP, which has been used to build the P300 speller, a communication tool through which users can type messages by controlling their eye-gazing [213]. Although P300 ERPs are collected through signal acquisition methods, these signals have very low signal-to-noise ratios; thus, the stimuli process must be continued to ensure the improved signal-to-noise ratios. This repetition has drawbacks, such as decreasing typing speed and increasing typing error [213]. In the reviewed literature, several supervised ML and DL algorithms have been applied to classify P300 responses correctly with a smaller number of repetitions. Among supervised ML algorithms, SVM and LDA have been applied successfully in P300 classification [125,214]. Furthermore, CNN with various architectures has been widely used in the P300 classification task [213,215,216,217]. However, the details on P300 classification using supervised ML and DL algorithms are beyond the scope of this review. In this section, we provide more comprehensive explanations of the contents of the articles reviewed. These explanations include further discussion of the applications of ML and DL algorithms and the feature extraction methods provided to ML classifiers depending on task types. This section also provides recommendations for selecting a ML and DL classifier according to the selected models in the reviewed articles, according to their observed performance.

### 5.1. Emotion Recognition Task

Emotions play essential roles in the evaluation of human behavior. EEG signals provide a convenient way to analyze an individual’s emotional response to external stimuli. Research has focused on classifying and predicting emotion dimensions while participants perform in externally driven activities, including watching video clips, facial pictures, and sequences of images [65,218,219]. In addition to exposure to these external stimuli, expression of the basic innate emotions (i.e., fear, anger, surprise, sadness, happiness, and disgust) causes different behavioral responses in individuals [220,221], Notably, fearful facial expression is a threatening stimulus resulting in an appropriate organization of defensive responses [222]. Moreover, environmental factors, such as approaching direction [223] have contributed to clarifying the effects of behavioral responses (i.e., defensive responses).

ER research based on EEG signals has recently become a commonly investigated topic (Figure 5). Of the 19 selected articles on this topic, 12 studies (63%) were published in the past 3 years. This field is expected to continue to grow over the next several years. Most studies used the Database for Emotion Analysis Using Physiologic Signals (DEAP), the most commonly used data to evaluate ER tasks and classify emotional states in two dimensions [161]. Table 3 briefly compares the performance of ML and DL algorithms observed in ER studies. As shown in Table 3, the NB classifier did not perform well for this task. Chung and Yoon [31] have used a weighted-log-posterior function for the Bayes classifier and reported an accuracy for valence and arousal classification of 66.6% and 66.4% for two classes and 53.4% and 51.0% for three classes, respectively. Shukla and Chaurasiya [167] have classified emotional states by using emotional dimension in the valence-arousal plane and reported that KNN outperformed other classifiers with an average accuracy of 87.1%. Three studies have analyzed differences in the performance of PCA in extracting desired features with different ML and DL classifiers. An earlier publication [164] has evaluated classification results both with and without a dimensional reduction technique; the results confirmed that KNN with PCA achieves better classification accuracy than other algorithms including SVM, LDA, LR, and DT. Bazgir et al. [163] have performed a similar analysis and decomposed EEG signals into five specific frequency bands by using discrete wavelet transform (DWT) via the db4 mother wavelet function on the selected channels. PCA was then applied to extract spectral features, and SVM with an RBF kernel was used to extract features from ten channels. This classification method achieved 91.3% and 91.1% accuracy for arousal and valence, respectively. Similarly, the high accuracy of ANNs has reported been in a previous study [166] using kernel PCA to extract segment-level features. Two additional studies have reported a high accuracy rate when using the RF algorithm. In one of these studies [162], the authors examined the level of fear based on emotional dimensions. They assessed fear in two- and four-level modes and built classifier models both with and without feature selection. The classification results confirmed that, use of differential entropy features resulted in the highest accuracy (90.07%) of RF classifiers in evaluating fear. Ramzan and Dawn [119] have observed similar results and reported that RF consistently outperformed other algorithms with an accuracy of 98.2% when its statistical features were used. Moreover, Nawaz et al. [168] have evaluated the performance of SVM, KNN, and DT classifiers with different types of features and found that statistical features performed best in assessing emotional dynamics in the human brain. Specifically, they achieved 77.62%, 78.96%, and 77.6% accuracy for binary classification of valence, arousal, and dominance, respectively, when these features were examined by using the SVM classifier. Qiao et al. [224] have proposed a novel model for multi-subject emotion classification. High-level features obtained by using the DL model and the CNN for feature abstraction resulted in a high accuracy of 87.27%. Overall, in studies on the DEAP data set, CNN can be chosen as a method to achieve competitive performance in ER tasks.

In contrast, the studies that did not use this shared data set [65,218,219,225,226,227,228,229,230] all achieved higher classification performance by using SVM, KNN, and ANNs algorithms. For example, Seo et al. [227] have compared different ML classifiers to classify boredom and non-boredom in 28 participants on the basis of the historical models of emotion and found that the KNN outperformed both RF and ANNs. Likewise, nearly identical performance has been reported by Heraz et al. [218] in experiments analyzing EEG signals from 17 participants. Murugappan [65] has reported a high accuracy performance for KNN used together with DWT. SVM is also a commonly used ML classifier; Li and Lu [219] have achieved the highest accuracy rate of 93.5% by using this method. In that study, participants were provoked with pictures of facial expressions. The authors applied CSP and linear-SVM and found that the gamma band (approximately 30–100 Hz) was suitable for classifying EEG-based human emotion. Additionally, in prior studies [27,230], tuned Q wavelet transform (TQWT) has been implemented to elicit the relevant features from the sub-bands. In reference [27], the classification of six extracted features and a probabilistic neural network (PNN) resulted in accuracy of 96.16% and 93.88% for classifying emotions in participants diagnosed with PD and healthy control subjects, respectively. Similarly, reference [230] has applied a multiclass least-squares support vector machine and achieved 95.7% accuracy for the classification of four emotions (happiness, fear, sadness, and relaxation). Because the performance of these classifiers differed among these nine publications, further investigation will be needed to identify the most effective ML classification algorithm. However, ANNs, KNN, and SVM methods with different kernels appeared to outperform other algorithms in ER tasks.

### 5.2. Mental Workload Task

Industrial sectors, including transportation, military, and aviation, require individuals to perform multiple tasks simultaneously (i.e., multitasking); operators in these settings must maintain constant vigilance to perform various tasks efficiently and effectively. MWL involves human factors that determine what resources may be required to perform a specific task [211]. Moreover, prior studies have accepted the role of human mental activities (i.e., human behavior) in human cognitive abilities such as working memory [231,232,233]. In this respect, Garofalo et al. [234] have indicated that individual differences are highly associated with the human behavior required by the task. Likewise, the influence of action control on execution and inhibition in motor responses in humans, and emotional stimuli in learned action have been discussed in references [235,236], respectively.

In the literature, three main approaches have been used to infer MWL levels: subjective measures, performance-based measures, and physiological measures [32]. Among these, approaches involving physiological methods that interpret MWL by using invasive, semi-invasive, and non-invasive physiological techniques perform relatively better than other measures. EEG data recording is a non-invasive method that provides a superior means to capture brain signals while participants perform complex tasks such as simulated driving or arithmetic calculations [32,237,238]. Overall, our review identified 23 publications that considered the MWL task. Different feature extraction techniques were widely applied in studies such as FFT, PCA, WT, AR, EMD, and HHT. Our investigation revealed that FFT, PCA, WT, and AR were the most widely used and effective methods for feature extraction among the reviewed articles in this domain; these methods were reported as superior 31%, 19%, 15%, and 15% of the time, respectively. For example, FFT has been implemented by researchers in several published studies [81,196,211,239,240,241,242] to extract features on the basis of the frequency of the EEG signals; however, another study has confirmed AR as one of the most reliable methods [243]. We also identified several essential feature extraction techniques that were less frequently applied in MWL tasks, including entropy and HHT, which elicited features of both nonlinear and non-stationary signals. Peng et al. [84] have applied HHT and SVM to evaluate attentiveness levels, and Vanitha and Krishnan [244] have used the HHT algorithm to extract EEG features to detect student stress levels. The study in reference [245] used the statistical method known as approximate entropy; more details on this method are provided in reference [246].

Various classifiers have been applied to identify the patterns in mental tasks to enhance classification performance. Our review indicated that SVM was the most commonly used ML algorithm to model MWL tasks. However, other ML methods, including KNN, ANNs, RF, and XGBoost classifiers have been used by others. For example, Dempster-Shafer theory and KNN (DSTKNN) classification methods have been applied in reference [243], in which desired features from EEG signals extracted by using both the AR model and statistical wavelet decomposition were provided to the DSTKNN classifier. The proposed algorithm achieved higher accuracy (93%) than did simple KNN. Likewise, Shah and Ghosh [55] have developed a real-time classification system by using PCA and a simple KNN classification algorithm. Interestingly, the studies in references [196,241] have proposed using FFT to evaluate the PSD on the basis of the time domain features incorporated in different ML models for classification. The results indicated the highest accuracy with KNN, at 99.42% and 90.5%, respectively. In addition to KNN, the application of ANNs [81], RF [211,240], and XGBoost classifiers [32] has been reported to identify individual intelligence quotients, mental work estimation, mental state, and task complexity. DL-based models have been applied in MWL tasks and have shown competitive performance in classification. For example, Jiao et al. [242] have designed a novel CNN architecture with an average result accuracy of 90% in 15 participants. In reference [247], the MWL has been classified by using the KNN classifier, the LSTM classifier, and the CNN + LSTM network. The best performance (61.08%) was achieved for the LSTM classifier. Given the proposed ML and DL algorithms, the FFT model has been implemented in several studies [81,211,240,242] to extract frequency domain features from different EEG bands.

Several studies have indicated that SVM classifiers are among the best techniques. Table 4 illustrates the highest classification accuracy of SVM when used with different types of feature extraction methods. In each study, the number of participants was six to eight, except for one previous study [84] that included 20 participants. After extraction, the combinations of all features were classified. As shown in Table 4, entropy, FFT, and EMD were associated with the highest accuracy when used with the SVM algorithm. The classification accuracy of entropy used with Immune Feature Weight SVM (IFWSVM), FFT with cubic SVM, and EMD with SVM were 97.5%, 95%, and 94.3%, respectively.

Seven types of ML and DL classification algorithms were used to study MWL tasks: CNN, LSTM, SVM, KNN, ANNs, RF, and XGBoost. Two studies [196,241] have used KNN classifiers with the feature extraction method FFT and compared the performance of KNN versus other algorithms. Both studies have reported that KNN with FFT results in the most accurate performance. Moreover, the studies in references [245,251,253] have reported that IFWSVM and RBF-SVM are more efficient than simple SVM at this task. Guo et al. [245] have proposed the IFWSVM classifier. The immune algorithm was applied on the basis of the assumption that each feature contributes differently to the overall result. Optimal feature weights were extracted with the immune algorithm and used to train the IFWSVM classifier. A previous study [251] has compared the performance of different feature extraction algorithms and found that an AR model-based representation performs better than frequency-based representations. The remaining studies have used a variety of other feature extraction methods. On the basis of our finding, CNN, SVM (particularly IFWSVM and cubic SVM) and KNN classifiers have been selected as good candidate classifiers for tasks associated with the MWL.

### 5.3. Motor Imagery Task

BCI allows for communication between users and external devices by translating brain signals into computer commands [68,254]. Recently, BCI technology has made significant contributions to neuroscience by enabling analysis of brain signals with high temporal resolution [88,254]. The main EEG paradigms used by BCI systems include ERP [255], SSVEP [256], and MI. MI is the mental performance of a movement without any evident muscle activation. However, MI leads to the activation of specific brain areas in a manner similar to that observed in association with actual muscle movement. In MI tasks, EEG signals are recorded while participants are asked to imagine certain muscle movements of their limbs [257]. MI can be used as a component of a treatment plan to promote recovery in patients with limited motor abilities. Because of the complexity of the brain, various methods have been used for feature extraction to discriminate between features of MI tasks. The selected features were provided to classification algorithms that were asked to differentiate between MI tasks including left- and right-handed movements and movements of a foot or a limb. Of the 26 publications focused on MI tasks, the best performance was reported in studies using DL such as CNN, and ML such as SVM, LDA, LR, RF, and NB classifiers.

As reported earlier [68,176], CSP is the most common feature extraction method and has been used extensively in BCI tasks, in agreement with our findings. Of the 26 selected studies focusing on MI tasks, ten (38%) applied this method with various classifiers. For example, Islam Molla et al. [176] have used CSP to extract four sets of features from each sub-band; the selected features were then provided to the SVM classifier, thus resulting in nearly 93% accuracy. Similarly, CSP has been applied in the studies featured in references [53,171] together with an SVM classifier. Soman et al. [171] have proposed a robust classifier using Twin-SVM and reported that this method enhanced the classification performance of left and right limb movements in the BCI Competition data set by 20%. However, CSP does not always perform perfectly with the SVM algorithm. Earlier studies [172,175,258] have compared the performance of ANNs and LDA with SVM. Jia et al. [175] have compared the results obtained by using backpropagation neural network (BPNN) and SVM algorithms and found that BPNN with CSP consistently outperformed SVM. The accuracy rates achieved with BPNN and SVM classifiers were 91.6% and 88.8%, respectively. Aljalal and Djemal [172] have implemented three classifiers using both CSP and WT. The authors found that LDA was more powerful with the features obtained from CSP and resulted in an accuracy rate of 84.79%; in contrast, SVM together with the WT method, achieved only 82.64% accuracy.

Two additional studies [34,124] have examined the performance of CSP and LDA for feature extraction and classification, respectively. In reference [124], the results indicated an 89.84% classification accuracy with the BCI-Competition IV data set. Similarly, a previous report [34] has indicated a 74.69% classification accuracy with EEG brain signals recorded from eight pianists. In addition to CSP, WT is a powerful tool that can be used to extract desired features in MI tasks. Sreeja et al. [61] have assessed the performance of the NB algorithm with WT and found a much higher classification accuracy than those obtained with either LDA or SVM classifiers. Ines et al. [174] and Maswanganyi et al. [52] have performed similar experiments and analyzed performance differences in ML classifiers by using the BCI Competition data set. Both studies have reported high accuracy when using WT with SVM and WT with NB, respectively.

One group of researchers has implemented three classifiers (SVM, LR, and KNN) to distinguish real from imagined movements by using EEG signals [195]. In this case, the LR algorithm outperformed both SVM and KNN, with overall accuracy rates ranging from 37% to 90%. Behri et al. [104] have compared several algorithms that might be used to differentiate EEG signals from the right foot and the right hand in five study participants. WPD was used to extract the features, and the RF classifier and WPD achieved a maximum accuracy of 98.45% in all participants. Similarly, Attallah et al. [259] have applied four levels of WPD to decompose EEG signals. Four different features were extracted, which were then introduced into the ML algorithm. With the SVM classifier, the highest accuracy of 93.46% and 86% was achieved for the BCI Competition III-IVa data set and the autocalibration and recurrent adaptation data set, respectively. Furthermore, a multi-class Adaboost-ELM algorithm based on features extracted by Kc has been used to classify three states (i.e., left-hand movement, right-hand movement, and resting-state) in ten participants [210]. Moreover, the performance of SVM was compared with that of Fisher linear discriminant, BPNN, and radial basis function neural network (RBF-NN) to achieve optimal performance of classification [260]. Yang et al. [261] have proposed a new framework based on multiple Riemannian covariances and MLP for feature extraction and classification, respectively. Use of this framework to classify MI EEG signals achieved a mean accuracy of 76%.

Several reviewed studies have used convolutional layers for EEG-based MI tasks. Sun et al. [262] have proposed the sparse spectrotemporal decomposition (SSD) algorithm for feature extraction and the CNN classifier with a 1-D convolution layer. Experimental results on BCI Competition IV and Tianjin University (TJU) data sets have achieved 79.3% and 85.7% accuracy, respectively. In reference [263], Liu et al. have considered the original EEG signals and their wavelet power spectrum as model input and reported that a deep CNN architecture based on space-time features and time-frequency features significantly improved the average accuracy performance of four-class MI classification. Dai et al. [264] have designed a hybrid architecture in which a convolutional layer CNN was used to learn network parameters, and the extracted features were then fed into the variational autoencoder (VAE) network.

Overall, we found no comprehensive comparisons of methods used as classifiers for MI tasks. Nevertheless, CNN, SVM, and NB outperformed other ML and DL classifiers in the articles reviewed in this study. On the basis of the findings, we suggest that CNN, SVM, and NB classifiers together with either CSP or WT extraction methods might be used for MI tasks.

### 5.4. Seizure Detection Task

Epilepsy is a neurological disorder whose diagnosis with automated SD has recently received significant attention [189], because the features extracted from the EEG signals are advantageous. In this task, EEG signals were recorded occasionally from healthy participants and patients with epileptic symptoms [20]. Seven ML and DL algorithms were used in SD studies: CNN [265,266], SVM [116,118,179,180,182,184,191,192,198,200,207,267], KNN [189,268,269], ANNs [183,199], RF [185,187,190], LDA [186], and ELM [181]. However, among the 24 studies focused on seizures, 12 applied the SVM algorithm with various kernels. Murugappan and Ramakrishnan [118] have used a hierarchical multi-class SVM (H-MSVM) with an ELM kernel to classify SD. Likewise, linear, and Gaussian kernels have been applied for high-dimensional spaces, respectively [198,267]. The RBF kernel is a common function used in the SVM algorithm applied to different tasks, as described by Hamed et al. [179] and Jaiswal and Banka [180]. Furthermore, the most common feature extraction methods reported in the publications selected for our study were WT [118,179,183,184,198,199,267], EMD [116,182,192], PCA [180,189,190,207], and FFT [268,269]. Similarly, the statistical feature extraction method 1D-TP used widely for image processing [270] has been applied [187] to generate the lower and upper features of each signal. Maximum difference of amplitude distribution histogram (MDADH) is a supervised feature selection method based on amplitude distribution histograms generated in both preictal and non-preictal trials. This method has been used by Bandarabadi et al. [200] to select and extract the desired EEG signal features. Further details on amplitude distribution histograms have been provided in previous publications [271,272]. The use of VMD, an extension of EMD, has been proposed by Chakraborty and Mitra [185]. The objective of VMD is to decompose input signals into sub-signals called modes. Because of the difficulties involved in selecting the appropriate number of modes and the associated penalty coefficient, the authors have proposed a kurtosis-based method that can be used to select the optimal number of modes and the penalty coefficient. More details on the VMD method can be found in reference [83].

As documented in Table 5, three shared SD databases have been used in the studies featured in this review: the BONN database, the CHB-MIT database, and the European epilepsy database. Some studies using the BONN database achieved near-perfect classification accuracy, reaching 100% with an SVM classifier [179,180,184] or RF classifier [185], and 97.7% accuracy with CNN [265,266]. In recent studies, residual CNN models using raw EEG signals as input [265,266] have also achieved competitive performance. In reference [180], feature extraction has been performed by using both sub-pattern-based PCA (SpPCA) and cross-sub-pattern correlation-based PCA (SubXPCA) methods; the extracted features were then sent to an SVM with an RBF kernel for further analysis. The feature extraction methods DWT and WPD have been used in references [179,181], respectively. In studies using the CHB-MIT database [190,191], the classification performance has reached 97.12% sensitivity with RF with PCA as a feature extraction method and 96% sensitivity with SVM. In addition, the use of SVM and ANNs classifiers resulted in nearly identical performance in studies using the publicly available European epilepsy database [199,200], in which an average sensitivity value of 73.5% was achieved.

Because we observed only trivial performance differences when comparing studies performed on data from each of the shared databases, we concluded that further research is needed to identify the most effective ML and DL algorithms for seizure detection. Nevertheless, as shown in Table 5, both CNN and SVM are robust algorithms that can be used to detect abnormalities from biomedical signals, and RF and ANNs are the most commonly used classifiers used in the studies included herein that focused on SD tasks. Thus, we recommend CNN, SVM, ANNs, and RF as good candidate ML and DL classifiers for this type of task.

### 5.5. Sleep Stage Scoring Task

A sleep stage is a period of time in which the sleep process remains constant. Sleep researchers focus on two main stages of sleep: rapid eye movement (REM) and non-rapid eye movement (NREM) [273]. NREM is divided into four stages (stages 1–4), each of which has specific characteristics. EEG signals are recorded during sleep and scored by experts that can classify them into REM or one of the four NREM sleep stages by using various detection methods.

Several ML techniques for automated SS tasks have been applied in the articles included in this review (Table 6). Ebrahimi et al. [274] have identified four sleep stages, extracted features based on WT coefficients, and applied MLP with eight neurons in one hidden layer, achieving 93% accuracy. Similarly, Zoubek et al. [275] have compared the performance between FFT and WT with two different classifiers: KNN and MLP. High accuracy was achieved with MLP with six neurons in one hidden layer when FFT was used as the feature extraction method. In another study [276], the researchers obtained time- and frequency-domain features from polysomnography signals by using dendrogram-based SVM as the classifier. The specificity, sensitivity, and accuracy of the classification of five sleep stages reached 94%, 82%, and 92%, respectively. Correspondingly, dendrogram-based multi-class SVM and WT have been used in one study [202] to classify three sleep states (light sleep and REM, deep sleep, and the awake) with an accuracy of 91.4%. Kuo and Liang [277] have proposed multiscale permutation entropy analysis for sleep scoring tasks along with the AR model and LDA, and have achieved a sensitivity of 89.1% for ten participants. Similarly, Santaji and Desai [59] have compared the performance of three classifiers: RF, SVM, and DT. The RF algorithm, when trained with extracted statistical features outperformed the other two algorithms in classification, with high specificity, sensitivity, and accuracy of 96.35%, 96.12%, and 97.8%, respectively. Delimayanti et al. [204] have applied FFT to elicit high-dimensional features and enhanced the classification performance of SS tasks by using the SVM algorithm with RBF kernel, and have achieved an average accuracy of 87.84%. Moreover, the performance of the AdaBoost classifier has been analyzed in reference [203,278]. Hassan and Bhuiyan [203] extracted time-frequency features by using the EMD method and compared the performance of different ML classifiers for SS task on the basis of a single channel EEG signal. AdaBoost significantly outperformed the other algorithms, with an accuracy of 92.24%.

### 5.6. Neurodegenerative Disease Task

Chronic pain is a brain disease that often occurs in older people [280] with a diagnosis of ND, such as AD, PD, and SZ [281]. According to reference [280], chronic pain in older patients may reduce memory extinction and increase the resilience of pain memory, as discussed by Battaglia et al. [282], who have demonstrated that older individuals have reduced extinction of fear memories.Analysis of EEG is a well-established method that can be used to detect brain abnormalities associated with these diseases. To perform this task, EEG data were recorded for several hours from both healthy participants and patients with these disorders to create a large data set. Analysis of brain signals to diagnose ND has been proposed in only a small number of selected studies. However, ML and DL applications offer new and potentially highly accurate approaches that might be used to diagnose brain abnormalities during early stages of the disease.

In the eight articles on AD selected for this review, the number of participants per study varied between 35 and 189, and the samples of patients with AD, patients with Mild Cognitive Impairment (MCI), and Healthy Controls (HCs) were well balanced. Four of these studies [117,283,284,285] explored the differences between patients with AD and HCs, whereas the other publications [58,102,274,286] considered all three groups. EEG band-pass filtering is a common strategy used to improve the spectral components of EEG signals; five of the eight publications have reported an EEG bandwidth at or below 40 Hz. The publications included in this review used three EEG feature extraction methods—spectral entropy, FFT, and WT—for the characterization of AD. As reported by Kulkarni [283,284], the combination of spectral entropy and an SVM classifier, compared with that of KNN classifier, resulted in outstanding performance achieving an accuracy of 96%. Two groups have used FFT to extract the EEG features [117,286]. Fiscon et al. [286] have applied FFT to the EEG signals and compared the outcomes associated with various classifiers, including SVM, DT, and rule-based classifiers. The results revealed that the use of DT to classify MCI versus HCs, AD versus MCI, and AD versus HCs was superior to the use of other classifiers, and achieved an accuracy of 90%, 80%, and 73%, respectively. Likewise, FFT has been used in one study [117] together with SVM to differentiate patients with AD from HCs, with an accuracy of 87%. WT has been developed into discrete and continuous WT feature extraction methods. The combination of DWT with DT has been examined in reference [102]. In addition, in reference [58], continuous wavelet transform (CWT) with MLP achieved the best accuracy, at 95.76% and 86.84% for AD versus HCs and AD versus MCI, respectively. Oltu et al. [274] have proposed an algorithm including three main steps that can be used to elicit the relevant features: DWT to extract EEG sub-bands, Burg’s method to measure the PSD of each sub-band, and a means for determining the amplitude summation of the coherence values for sub-bands. In this study, Bagged Trees outperformed the other classifiers, with a classification accuracy of 96.5%. CNN architecture, used by Morabito et al. [285] with two hidden convolutional layers to extract features of multi-channel EEG signals, has been reported to achieve an 82% accuracy for three-class classification.

Two PD-related articles were considered in the ND category. These studies were performed on experimental data sets with 20 or 18 participants, in references [209,287], respectively. According to the study published in reference [287], FFT was used as a feature extraction method together with a KNN classifier to differentiate between PD patients and HCs, with 88% accuracy. HOS, a powerful method to extract the nonlinear EEG signal features [288], was introduced by Yuvaraj et al. [209]. In that study, the use of RBF-SVM resulted in an accuracy of 99.62%.

Only a very small fraction of the publications included in this review were SZ-related studies. In one study [212], wavelet-based features were elicited from a single channel and classified with the KNN algorithm. Classification accuracy values of 99.21% and 97.2% were obtained with 10-fold and leave-one-subject-out cross-validation methods, respectively. Other researchers have designed RF classifiers by using features extracted based on ERP [289]. With 10-fold cross-validation, the best classification accuracy of 81.10% was achieved.

## 6. Future Directions

Several promising future directions exist in the implementation of EEG signal classification. Most recent research has focused on using DL algorithms that require increasing the amount of data and changing the structure of the model [290]. Although DL models can effectively solve the EEG signal classification tasks, transfer learning strategy from one model to another accelerates training time and yields the best performance results [291]. Another exciting prospect is applying a graph neural network (GNN) framework to consider the brain connectivity network by identifying Regions of Interests [292]. DL models cannot directly work on graph-structured input data because they consider the brain network features as a vector of one dimension [293]. The human brain connectivity represents the brain as a graph with interacting nodes in non-Euclidean space, and existing DL methods generally disregard the interaction and association of brain connectivity networks [294]. GNNs aim to learn graph representations by using a neural network and to pass information via a message-passing algorithm [295,296]. Unlike neural networks, GNNs update the representations of nodes while maintaining the graph topology.

## 7. Conclusions

On the basis of our review of 128 published articles, various supervised machine learning and deep learning algorithms have been widely applied in various tasks, including ER, MWL, MI, SD, SS, and diagnosis of ND. Several metrics can affect the performance of classifiers, including different data sets, preprocessing techniques, and feature extraction methods. We presented an overview of feature extraction methods as part of our findings addressing RQ2. We also introduced the publicly available databases that have frequently been used for each task, and we directly analyzed the classification performance reported in relevant studies. Many of the reviewed studies have compared the performance of different classifiers; CNN, SVM, and KNN were the most frequently used classifiers across all articles reviewed. Although model performance can be attributed to a variety of factors, our findings suggested that SVM and KNN outperformed the other supervised ML classifiers. We also found that CNN and NB had impressive performance in studies focused on MI tasks when either CSP or WT was used as a feature extraction methods. Similarly, the performance of RF was superior to that of the other classifiers in studies focused on ER tasks with the DEAP database. This systematic review provided recommendations for applying supervised machine learning and deep learning algorithms for the neural decoding of EEG signals in various tasks and experimental protocols. Although each classification algorithm has its own strengths and limitations, these recommendations provide insight into the issues associated with the classification of EEG signals, which might be addressed in future research efforts in this field. Further in-depth studies combining the selection of feature extraction methods and types of classifiers are highly recommended.

## Figures and Tables

**Figure 1 brainsci-11-01525-f001:**
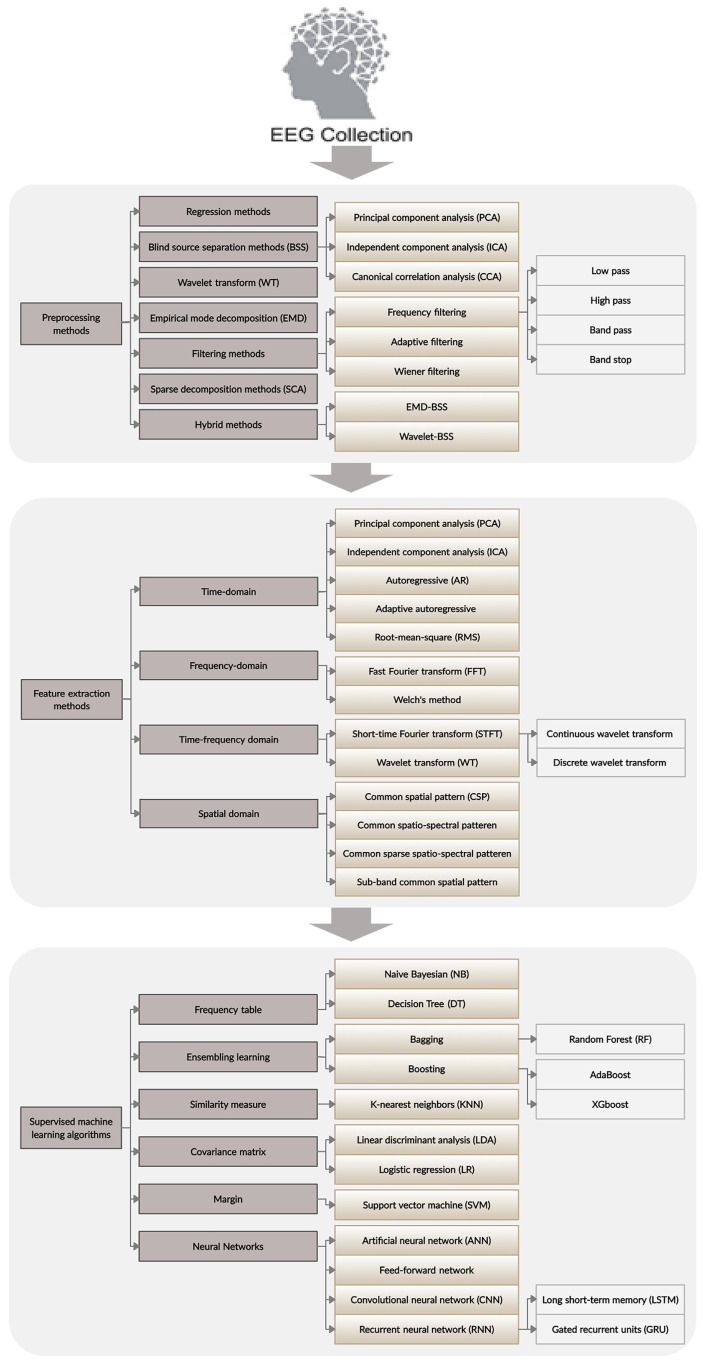
Flow chart of the EEG data analysis pipeline along with taxonomy of existing methods for each step.

**Figure 2 brainsci-11-01525-f002:**
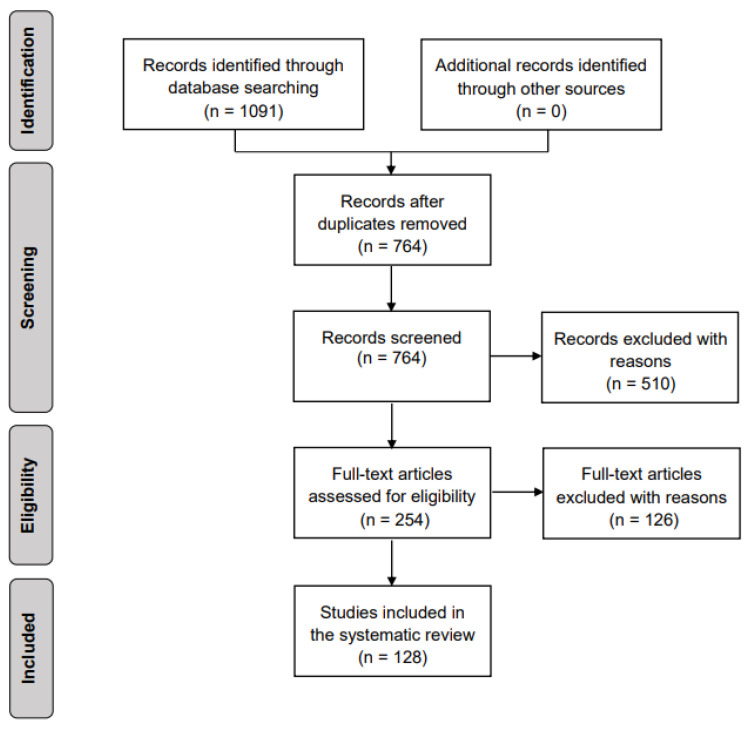
Flow diagram based on the PRISMA guidelines [38]. The diagram includes the four stages of a PRISMA study: identification, screening, eligibility, and inclusion.

**Figure 3 brainsci-11-01525-f003:**
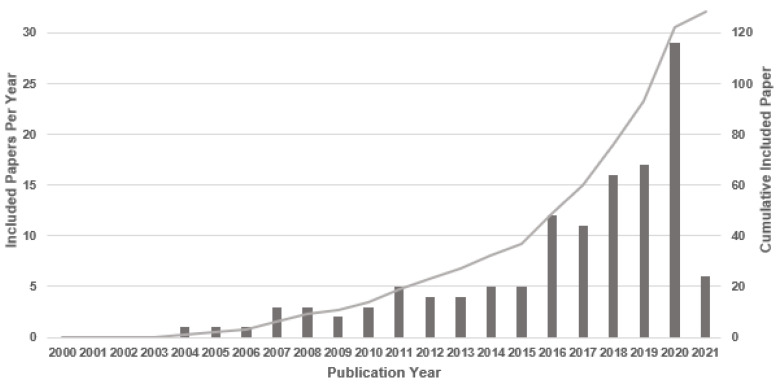
Temporal distribution of articles selected for consideration.

**Figure 4 brainsci-11-01525-f004:**
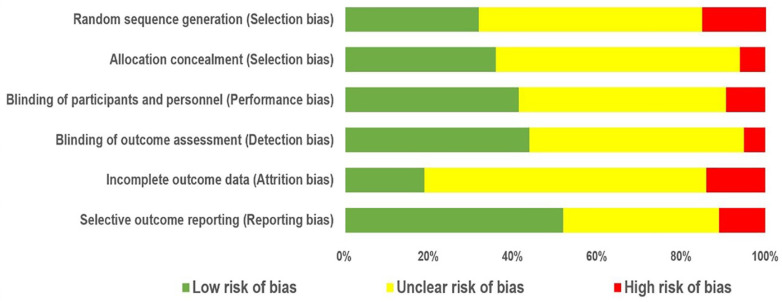
Assessment of the risk of bias in the 128 studies selected for this review by using the Cochrane Collaboration tool.

**Figure 5 brainsci-11-01525-f005:**
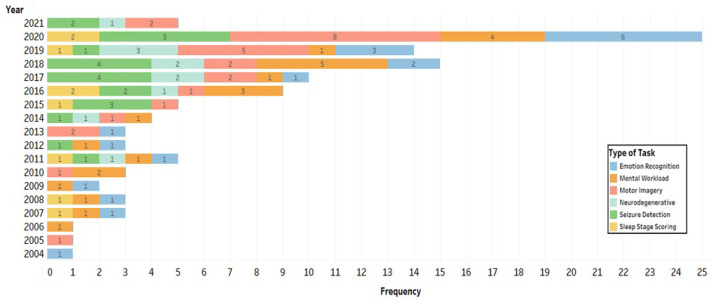
Temporal distribution of the number of publications per domain in each year (2004–2021).

**Figure 6 brainsci-11-01525-f006:**
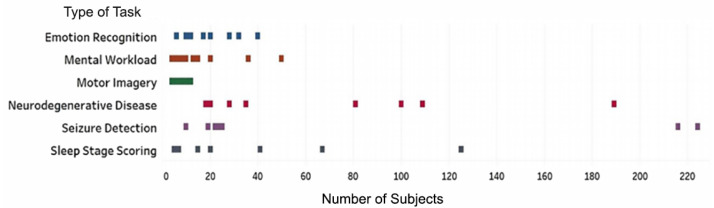
Number of subjects per task in each study reviewed.

**Figure 7 brainsci-11-01525-f007:**
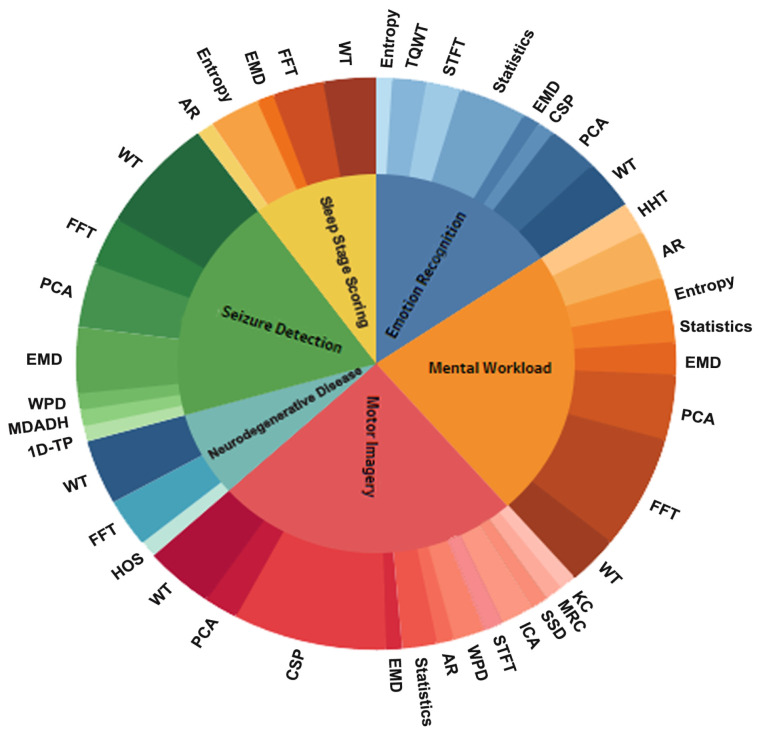
Feature extraction methods used in all studies considered in this review. The inner circle represents the type of task, and the outer circle represents the utilization rate of the method for each task. Abbreviations: AR, autoregressive; CSP, common spatial pattern; EMD, empirical mode decomposition; FFT, fast Fourier transform; HHT, Hilbert Huang transform; HOS, higher-order spectra; ICA, independent component analysis; Kc, Kolmogorov complex; MDADH, maximum difference of amplitude distribution histogram; MRC, multiple Riemannian covariance; PCA, principal component analysis; STFT, short-time Fourier transform; SSD, sparse spectrotemporal decomposition; TQWT, tunable Q wavelet transform; WPD, wavelet packet decomposition; WT, wavelet transform; 1D-TP, one-dimensional ternary patterns.

**Figure 8 brainsci-11-01525-f008:**
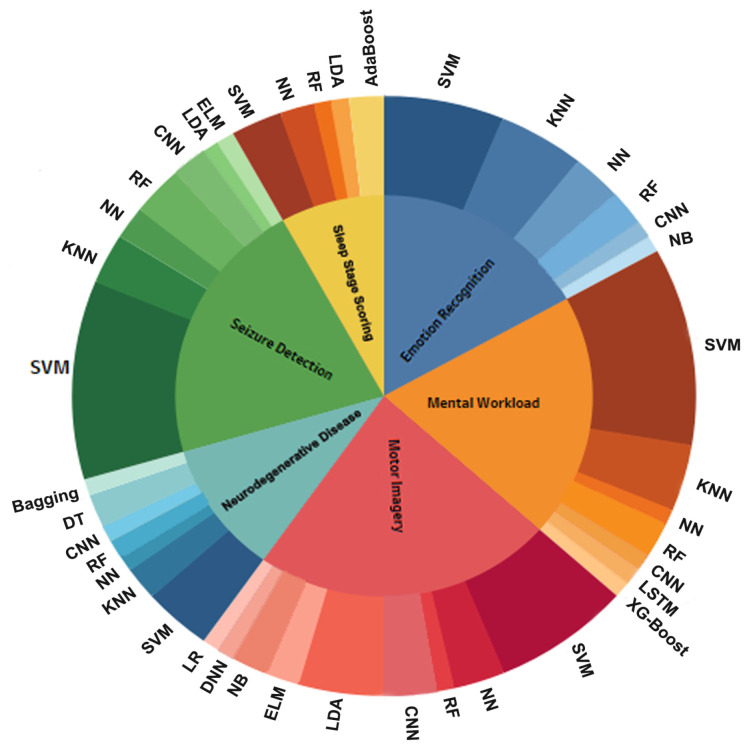
The most efficient machine learning algorithms used for different tasks. The inner circle represents the type of tasks, and the outer circle represents the utilization rate of the supervised machine learning and deep learning classification algorithms used for each task across all studies evaluated. Abbreviations: CNN, convolutional neural network; DNN, deep neural network; DT, decision tree; ELM, extreme learning machine; KNN, K-nearest neighbor; LDA, linear discriminant analysis; LR, logistic regression; LSTM, long short-term memory; NB, naïve Bayes; NN, neural networks; RF, random forest; SVM, support vector machine.

**Figure 9 brainsci-11-01525-f009:**
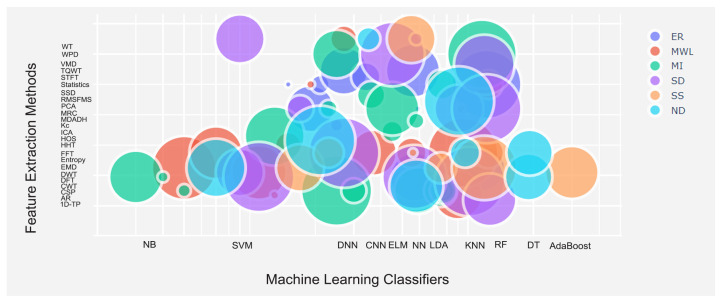
Bubble plot of studies according to classification algorithms and feature extraction methods. The size of the bubble indicates the performance of classification models for each task as marked with different colors.

**Table 1 brainsci-11-01525-t001:** Brief comparison of conventional classification algorithms and deep learning algorithms.

	Conventional Classification Algorithms	Deep Learning Algorithms
Input features	Hand-crafted	Automatically based on representation learning
Feature selection process	Required	Not required
Model architecture	Based on statistical concepts	Consists of a diverse set of architecture based on sample data
Computational cost	Computational cost is based on the conventional classification models but is lower than that of deep learning algorithms	Computational cost is very high, because hyper parameters must be tuned

**Table 2 brainsci-11-01525-t002:** List of public data sets used for the analysis of EEGs associated with different types of tasks.

Source	Database	Studies Using This Data Set	Number of Subjects	Target Tasks
Koelstra et al. [161]	DEAP	[31,119,162,163,164,165,166,167,168]	32	Emotion recognition
Blankertz et al., Leeb et al. [169,170]	BCI Competition	[49,61,104,171,172,173,174,175,176,177]	30 subjects in 4 different data sets	Motor imagery
Andrzejak et al. [178]	BONN	[118,179,180,181,182,183,184,185,186,187]	5	Seizure detection
Moody et al. [188]	CHB-MIT	[189,190,191,192]	22	Seizure detection
Goldberger et al. [193]	PhysioNet	[116,194,195,196]	109	Motor imagery/mental workload
Ihle et al. [197]	European Epilepsy	[198,199,200]	300	Seizure detection
Kemp et al. [201]	Sleep-EDF	[59,202,203,204]	197	Sleep stage scoring
Ichimaru et al. [205]	MIT-BIH	[159]	16	Sleep apnea detection
Winterhalder et al. [206]	Freiburg	[207]	21	Seizure detection

**Table 3 brainsci-11-01525-t003:** Comparison of studies conducted on emotion recognition tasks by using the DEAP data set.

Authors	Year	Feature Extraction Method	Classification	Performance (%)
Ramzan and Dawn [119]	2019	Statistics	RF	Accuracy = 98.2
Bazgir et al. [163]	2018	PCA	RBF-SVM	Accuracy = 91.1 (valence) Accuracy = 91.3 (arousal)
Balan et al. [162]	2019	Entropy	RF	Accuracy = 90.07
Qiao et al. [224]	2017	STFT	CNN	Accuracy = 87.27
Shukla and Chaurasiya [167]	2018	DWT	KNN	Accuracy = 87.1
Nawaz et al. [168]	2020	Statistics	SVM	Accuracy = 77.62 (valence) Accuracy = 78.96 (arousal) Accuracy = 77.6 (dominance)
Doma and Pirouz [164]	2020	PCA	KNN	Accuracy = 74.25
Chung and Yoon [31]	2012	N/A	NB	Accuracy = 66.6 (valence) Accuracy = 66.4 (arousal)
Rozgic et al. [166]	2013	PCA	ANNs	Accuracy > 60

**Table 4 brainsci-11-01525-t004:** Classification accuracy of the selected feature extraction methods with the SVM classifier for a mental workload task.

Authors	Year	Feature Extraction Method	Classification	Performance (%)
Guo et al. [245]	2010	Entropy	IFWSVM	Accuracy = 97.5
Rashid et al. [239]	2018	FFT	Cubic SVM	Accuracy = 95
Gupta and Agrawal [248]	2012	EMD	SVM	Accuracy = 94.3
Vanitha and Krishnan [244]	2016	HHT	SVM	Accuracy = 89.07
Wei et al. [249]	2011	PCA	SVR	Accuracy = 85.92
Peng et al. [84]	2020	HHT	SVM	Accuracy = 84.8
Gupta et al. [250]	2020	WT/EMD	Non-linear SVM	Accuracy = 80–100
Hosni et al. [251]	2017	AR	RBF-SVM	Accuracy = 70
Liang et al. [252]	2006	AR	SVM	Accuracy = 67.57

**Table 5 brainsci-11-01525-t005:** Comparison of studies conducted for the seizure detection task by using shared data sets.

Database	Authors	Year	Feature Extraction Method	Classification	Performance (%)
BONN	Hamed et al. [179]	2018	DWT	RBF-SVM	Accuracy = 100
	Savadkoohi et al. [184]	2020	FFT / WT	SVM	Accuracy = 100
	Jaiswal and Banka [180]	2018	PCA	RBF-SVM	Accuracy = 100
	Riaz et al. [182]	2015	EMD	SVM	High performance in the detection of seizures in case 1 and 2
	Ullah et al. [266]	2018	-	1D-CNN	Accuracy = 99.1
	Lu and Triesch [265]	2019	-	Residual CNN	Accuracy = 99
	Chakraborty and Mitra [185]	2021	VMD	RF	Accuracy = 98.7–100
	Ech-Choudany et al. [186]	2021	Dissimilarity-based TFD	LDA	Accuracy = 98
	Mardini et al. [183]	2020	DWT	ANNs	Accuracy = 97.8
	Liu et al. [181]	2017	WPD	ELM	Accuracy = 97.7
	Murugappan and Ramakrishnan [118]	2016	WT	H-MSVM	Accuracy = 94
	Kaya and Ertugrul [187]	2018	1D-TP	RF	Accuracy > 94
CHB-MIT	Pinto-Orellana and Cerqueira [190]	2016	PCA	RF	Sensitivity = 97.1 Specificity = 99.2
	Shoeb et al. [191]	2011	-	SVM	Sensitivity = 96
	Fergus et al. [189]	2015	PCA	KNN	Sensitivity = 93 Specificity = 94
	Usman et al. [192]	2017	EMD	SVM	Sensitivity = 92.2 Specificity = 93.4
European Epilepsy	Teixeira et al. [199]	2014	WT	ANNs	Sensitivity = 73.1
	Direito et al. [198]	2017	DWT	Linear-SVM	High performance in a small subset of participants
	Bandarabadi et al. [200]	2015	MDADH	SVM	Sensitivity = 73.98

**Table 6 brainsci-11-01525-t006:** Comparison of studies for the sleep stage scoring task, including sleep stages, feature extraction method, machine learning algorithm, and overall performance.

Authors	Year	Sleep Stages	Feature Extraction Method	Classification	Performance (%)
Santaji and Desai [59]	2020	S1, S2, REM	Entropy	RF	Accuracy = 97.8
Ebrahimi et al. [279]	2008	Awake, S1 and REM, S2, SWS	WT	MLP	Accuracy = 93
Hassan and Bhuiyan [203]	2016	Awake, S1, S2, S3, S4, REM	EMD	AdaBoost	Accuracy = 92.2
Lajnef et al. [276]	2015	Awake, S1, S2, SWS, REM	Entropy	Dendrogram-SVM	Accuracy = 92
Ravan [202]	2019	Awake, LS and REM, DS	WT	Dendrogram-SVM	Accuracy = 91.4
Kuo and Liang [277]	2011	Awake, S1, S2, SWS, REM	Entropy/AR	LDA	Sensitivity = 89.1
Delimayanti et al. [204]	2020	Awake, S1, S2, S3, S4, REM	FFT	RBF-SVM	Accuracy = 87.8
Zoubek et al. [275]	2007	Awake, NREM1, NREM2, SWS, PS	FFT	MLP	Accuracy = 71.6

## Data Availability

Not applied to this study.

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
