# Peer review of "Neural Decoding of EEG Signals with Machine Learning: A Systematic Review"

_brainsci, 2021, doi:10.3390/brainsci11111525_

Round 1
Reviewer 1 Report
The authors summarized the recent advances on neural decoding of EEG signals with machine learning. The review is a very meaningful and informative. But I find that there are some major issues with the paper that require addressing prior to this being considered for publication in this journal. This manuscript has some spelling typos, style errors and grammatical errors. Pleases carefully check and correct them in the revised manuscript. For a review manuscript, the detailed introduction of the relevant research progress is necessary, while the author's own views and objective evaluation are more important which are ignored, unfortunately. For emerging applications of EEG in dairy life, novel gel-free electrodes (including semi-dry and dry EEG electrodes) are often used to collect EEG signals. Due to the absence of conductive gels, the gel-free electrodes, especially for dry electrodes, have higher contact impedance, resulting in poor EEG signals. So, I am very interested in how machine learning can improve the classification accuracy of EEG signals recorded from such gel-free electrodes. I hope the author will add some relevant discussions in the revised manuscript. In addition, some related references are recommended to be cited, such as J. Neural Eng. 17 (2020) 026001; J. Neural Eng. 17 (2020) 051004; J. Neural Eng. 18 (2021) 046016; Sensors, 2014, 14, 12847-12870. In addition, the future direction should be pointed out in the end of this review. The length of this review is too long, please further refine and shorten the review.
Reviewer 2 Report
In the present review entitled ‘Neural Decoding of EEG Signals with Machine Learning: A Systematic Review’, by Saeidi and colleagues, authors aimed to investigate recent advancements in decoding EEG signals with machine learning and deep learning supervised models. For this purpose, they selected publications from multiple academic databases to find relevant studies from 2000 to the present on this subject. 128 articles met the inclusion criteria for the systematic review and were included in the synthesis. The results of this review provide recommendations for applying machine learning and deep learning algorithms that can be used for the neural decoding of EEG signals under different tasks and experimental protocols.
In general, I think the idea of the review is really interesting and the authors’ fascinating observations may be of interest to the readers of Brain Sciences. However, some comments, as well as some crucial citations that should be included to support the authors’ argumentation, need to be addressed to improve the review, its adequacy, and its readability prior to the publication in the present form.
Comments
- Regarding the abstract: according to the Journal’s guidelines, authors should have provided an abstract of about 200 words maximum. Indeed, the current one includes 268 words.
- Page 2, line 55-58: The statement “...binary and multi-label classifications are used widely in…studies of cognitive function, motor imagery (MI) processing, emotion recognition (ER), and brain disorders…” necessarily needs some citations. In this regard, I recommend crucial studies that could provide further insight on models contribute in the prediction and estimation of probable future outcomes: decisive evidence from Garofalo and colleagues’ study (2017, Journal of Cognitive Neuroscience) suggest that, according to the predicted response–outcome (PRO) model, medio-frontal ERP signals of prediction error track the timing of salient events and reports an error signal for outcomes occurring at unexpected times during a task with no action requirement. Also, a recent study by Battaglia and colleagues (2020, Journal of Neuroscience) might be of interest: in this study, authors showed the involvement of model-based computations in the acquisition of fear conditioning (i.e., learning) occurring in the ventromedial prefrontal cortex (vmPFC), that seems to be essential in the interpretation of actions’ value and stimulus-outcome contingencies. The role of vmPFC in model-based representations underlying fear learning, fear extinction or both is also been discussed in a recent review on vmPFC subregional contributions (Battaglia* et al., 2021, Molecular Psychiatry – In Press).
- Page 2, line 81: Please replace “provides” with “provide”.
- Page 7, line 243: Please replace “algorithms” with “algorithm”.
- Page 12, Deep Learning Algorithms: Authors explained how deep learning algorithms are suited to analyze data by storing information from previous output states. In my opinion, adding some references that explored how learning and memory processes can be modulated would be crucial in this section: for example, Borgomaneri and colleagues’ recent study (2020, Current Biology) shows that the inhibition of the dorsolateral prefrontal cortex (DLPFC) after memory reactivation disrupts physiological responding to learned fear, highlighting the role of this area in the neural network that mediates the reconsolidation of fear memories in humans. Similarly, I also suggest mentioning the recent review from the same research group (Borgomaneri et al., 2021, Neuroscience and Biobehavioral Reviews) the application of non-invasive brain simulation (NIBS) to modulate fear memories. Finally, I would suggest one of the latest Borgomaneri and colleagues’ study (2021, Journal of Affective Disorders), that illustrated the therapeutic potential of NIBS as a valid alternative in the treatment of abnormally persistent memories that characterized those patients with anxiety disorders that do not respond to psychotherapy and/or drug treatments. I believe that adding information regarding memory processing will dramatically improve the argumentation.
- Page 21, lines 651-655: when discussing the role that emotions have on human behavior, authors acknowledged that “research has focused on classifying and predicting emotion dimensions while subjects participate in externally driven activities, including watching video clips, facial pictures, and sequences of images”. According with this sentence, I would recommend a recent study by Ellena and colleagues’ study (2020, Experimental Brain Research), in which authors suggest that responses to images of approaching emotional stimuli can modulate autonomic arousal as a function of the distance between the observer and the stimuli, resulting in an appropriate organization of defensive responses. Similarly, results from a recent study by Candini and colleagues (2021, Scientific reports) show that interpretation of potentially threatening situations, such as stimuli approaching the subject, triggers a number of physiological responses that help regulating the distance between ourselves and others during social interaction. Furthermore, in a recent yet relevant study, Borgomaneri and colleagues (2021, Brain Sciences) investigated the impact that pictures of emotional stimuli, combined with transcranial magnetic stimulation (TMS), have on of corticospinal excitability.
- Page 21, lines 731-732: According to authors, “MWL involves human factors that indicates what resources may be required to perform a specific task”. In this respect, a study by Garofalo and colleagues (2019, Scientific reports) might be of interest: in this study, in which 100 healthy participants performed a decision-making and learning task, authors demonstrated that individual differences in cognitive abilities, such as working memory, are crucially related to the specific individuals’ behavior requested by the task and thus, guide behavior toward the most convenient choice. Adding to this evidence, in a recent review, Borgomaneri and colleagues (2020, Cortex) outlined neural circuits underlying action control, required for the execution and inhibition of motor responses in humans; the same research group (Battaglia et al., 2021, Behaviour Research and Therapy) also explored the impact that emotional stimuli have on action inhibition of ongoing learned actions.
- Page 27, Neurodegenerative Disease Task: Authors described how machine learning and deep learning algorithms that might be used to diagnose brain abnormalities during the early stages of the disease. In this regard, I suggest mentioning Battaglia and colleagues’ study (2018, Scientific reports), that has focused on age-related impairment in fear extinction, proving the influence of normal aging on reduced ability to use contextual information to modulate responses to threat. Accordingly, Cai and colleagues’ study (2019, Scientific reports) used a machine learning model to train a model for personalized forecasting of Alzheimer’s disease progression, providing the possibility to develop tools that can simulate patient progression in detail.
- Page 28, Conclusions: In this paragraph it would be really interesting if the authors discussed about their work of neural decoding of the EEG signal in addition to the new models of predictive coding, which at this moment is of interest across many emerging fields in neuroscience.
Round 2
Reviewer 1 Report
The information in Reference 40 is incomplete. Please correct it.